# The Role of Lipid Metabolic Reprogramming in the Hibernation of Chipmunks

**DOI:** 10.3390/ani15142091

**Published:** 2025-07-15

**Authors:** Mingrui Huang, Chong Wang, Juntao Liu, Qing Liu, Ye Tian, Xiaohui Li, Wei Lu, Dawei Zhang, Huimei Yu

**Affiliations:** 1Department of Pathophysiology, College of Basic Medical Sciences, Jilin University, Changchun 130021, China; huangmr2020@mails.jlu.edu.cn (M.H.); wangchong.e@163.com (C.W.); liuqing23@mails.jlu.edu.cn (Q.L.); tiany24@mails.jlu.edu.cn (Y.T.); lixiaohui@jlu.edu.cn (X.L.); dwzhang@jlu.edu.cn (D.Z.); 2School of Public Health, Jilin University, Changchun 130021, China; jtliu2720@mails.jlu.edu.cn; 3Animal Disease Prevention and Control Center of Chaoyang District, Changchun 130021, China; cccynongye@sohu.com

**Keywords:** hibernation, liver, lipid metabolism, transcriptome sequencing

## Abstract

Hibernation is a special physiological state that animals enter in response to cold stimuli. In this study, chipmunks were used to investigate the adaptive changes that occur in the liver during hibernation. Studies have shown that chipmunks have a compensatory increase in liver volume and a significant change in energy supply pattern from glucose metabolism to fat metabolism during hibernation. It is this change in energy metabolism that allows chipmunks to cope with the cold.

## 1. Introduction

Hibernating animals require a sufficient energy supply and fat. As they transition from an active state to hibernation, their energy source shifts from carbohydrates in their food to stored fat, a transition from high levels of glucose metabolism to lipid metabolism [1]. Eukaryotes store most of their energy in the form of lipids, which represent a long-term energy reserve [2], while carbohydrates are considered short-term energy reserves. Lipids are energy-dense molecules that have the greatest energy yield per unit weight and contribute greatly to energy homeostasis, temperature regulation, and membrane fluidity [3,4]. Fats, in the form of triglycerides, are the most energy-dense form of energy storage, which is why triglycerides have been selected as the main energy reservoir during evolution instead of other macromolecules [5].

The wild chipmunks in the northern regions of China enter the hibernation period around November, when the ambient temperature drops below 4 °C. They wake up around March of the following year. Chipmunks prepare for hibernation by the augmentation of fat and food reserves, wherein the former plays a dominant role. Unlike the hibernation patterns of some animals, chipmunks consume food and water intermittently during hibernation [6], and their thermoregulation relies on lowering their organic metabolic levels, including reduced activity, diminished muscle contractility, decreased brain function, and heat regulation processes [7] including dynamic adipose-tissue-derived non-shivering heat production from brown fat, resulting in heat redistribution across the body surface and the vital organs [8]. Lipid metabolism plays a crucial part in the whole process of hibernation.

However, the accumulation of sufficient fat is often considered a threat to bodily health, leading to numerous diseases such as insulin resistance, inflammation, and cardiovascular disease [9]. It is worth noting that most hibernating animals are completely free of these conditions, giving rise to the mystery of how they stay healthy. Hibernating animals can consume fat rapidly [10], but this metabolic change does not cause significant physical damage, which has aroused great scientific interest in the metabolic regulation of the liver in hibernating animals. The liver is important for glucose metabolism, lipid metabolism, and protein metabolism, and is a vital organ for energy supply. There are few studies on how the liver plays its regulatory role in hibernation.

For this research, we compared the effects of hibernation on the liver morphology of chipmunks and evaluated liver energy levels during hibernation. Subsequently, transcriptome sequencing was used to analyze the expression levels of the genes involved in glucose metabolism, lipid metabolism, and energy supply conversion in the livers of chipmunks. Through our study, we not only deeply analyze the impact of cold environments on body metabolism but also provide adequate strategies and powerful theoretical support for studies in the fitness, nutrition supplement, and training monitoring of diabetic people, obese people, winter sports enthusiasts, and winter sports athletes.

## 2. Materials and Methods

### 2.1. Animals and Study Design

In September 2022, 20 wild adult chipmunks were captured in Changchun City, Jilin Province, comprising an equal number of males and females. This research plan was approved by the Experimental Animal Ethics Committee of the School of Basic Medicine, Jilin University, and the use of wild chipmunks was also agreed upon by the relevant institutions. After capture, the wild chipmunks were acclimatized and fed in the biosafety level 2 laboratory of Jilin University for 14 days. The wild chipmunks were randomly divided into two groups, with 10 in each group and an equal number of males and females. One group was named the hibernation group (HIB) and the other the control group (CON). The hibernation group was placed in a constant-temperature box at (3 ± 1) °C without light, while the control group was placed at a room temperature of (22 ± 2) °C. Both groups of chipmunks were allowed free access to food and water. Starting from the first day of inducing hibernation, every 14 days, at 8:00 in the morning, blood samples were collected from the tail vein of the hibernating group and the control group of chipmunks. The blood samples were directly tested for blood sugar using glucose test strips. After 120 days, the chipmunks in both groups were euthanized using carbon dioxide inhalation. The specific method was as follows: pure carbon dioxide filled a chamber for 2 min, and, after waiting for 2 min, it was confirmed that the chipmunks had died. Then, the carbon dioxide was removed, and the chipmunks were taken out. The livers, hearts, kidneys, lungs, intestines, rectus femoris, and extensor hallucis longus of the chipmunks were collected on a super-clean workbench. Samples were immediately stored in liquid nitrogen and 4% paraformaldehyde fixation solution.

### 2.2. Histological Analysis of the Liver Tissues

The fixative solution was washed off the surface of the liver tissue. After dehydration through an ethanol series, the tissue was embedded in solid paraffin and then cut into 5 μm sections with a microtome. The slices were placed on slides and stained with hematoxylin-eosin (HE) before being observed using a microscope (Olympus BX53, Shinjuku, Tokyo, Japan).

### 2.3. Glycogen Content Detection

For this test, 50 mg of tissue (liver, heart, kidneys, lungs, intestines, rectus femoris, and extensor hallucis longus) from each chipmunk was weighed, and the extraction solution was added, then the mixture was boiled in boiling water for 20 min and thoroughly mixed. When the tissue had completely dissolved, the volume was fixed to 5 mL after cooling and mixing. The reagents were prepared according to the instructions for the glycogen detection kit (Solarbio, Beijing, China) and added to the EP tube in order. The spectrophotometer was adjusted to a wavelength of 620 nm for detection. Glycogen content was then calculated according to the mass of the tissue sample.

### 2.4. Blood Glucose Measurement

Affectionate Pet blood glucose test strips (C242739-C, VivaChek Inc., Hangzhou, China) were used to measure the blood glucose level of the newly collected blood from the chipmunks.

### 2.5. ATP Assay

The intracellular ATP levels of the liver tissues were determined using an enhanced ATP assay kit (Beyotime, Shanghai, China). Fifty milligrams of fresh animal tissue was weighed, and lysis buffer was added at a ratio of 200 μL per 20 mg of tissue. After thorough homogenization, the mixture was centrifuged at 12,000× *g* for 5 min at 4 °C, and the supernatant was collected for subsequent assays. In white opaque 96-well plates, the prepared ATP assay working solution and samples were added in the order specified by the ATP assay kit instructions. Three replicate wells and blank controls were set up, and relative light unit (RLU) values were measured using a luminometer. The standards provided in the ATP assay kit were used to generate a standard curve, based on which the ATP content in the tissues was calculated.

### 2.6. Fatty Acid Analysis

We homogenized 50 mg of liver in 300 μL of physiological saline, then added 1.2 mL of a chloroform: methanol (2:1, *v*/*v*) solution and incubated overnight at 4 °C for total lipid extraction. After centrifugation at 900 g for 5 min, we collected the lower layer of the solution in a new glass centrifuge tube. We added 50 μL of a 5 mg/mL C17:0 standard solution, mixed well at room temperature, then added 100 μL of sodium methoxide solution, which was mixed for an additional 5 min to perform the derivatization reaction. We quenched the reaction by adding 300 μL of 3N methanolic hydrochloric acid, followed by the addition of 600 μL of n-hexane to extract the FAMEs (fatty acid methyl esters). We collected the upper n-hexane phase, dried it under a stream of nitrogen gas, and redissolved the residue in 50 μL of n-hexane containing 2 g/L butylated hydroxytoluene (BHT). Finally, we injected 1 μL of the sample for GC (gas chromatography) analysis.

### 2.7. Transcriptome Sequencing

Total RNA was isolated from chipmunk liver samples using the TRIzol reagent (Invitrogen, Carlsbad, CA, USA). For RNA-sequencing (RNA-seq, Biomarker Technologies, Beijing, China) analysis, three animals per genotype, Insm1 1/lacZ (control), Insm1 lacZ/lacZ (Insm1 null), and Insm1 delSNAG/lacZ (Insm1del-SNAG), were collected. Sequencing libraries were generated using the NEBNext Ultra Directional RNA Library Prep Kit for Illumina (Illumina, San Diego, CA, USA). The clustering of the index-coded samples was performed on a cBot Cluster Generation System using TruSeq PE Cluster Kit v3-cBot-HS (Illumina, San Diego, CA, USA) according to the manufacturer’s instructions. After cluster generation, the library preparations were sequenced on an Illumina HiSeq 2000 platform (Illumina, San Diego, CA, USA) and paired-end reads were generated. Raw data (raw reads) in the fastq format were first processed through in-house Perl scripts. In this step, clean data (clean reads) were obtained by removing reads containing the adapter, reads containing ploy-N, and low-quality reads from raw data. At the same time, the Q20, Q30, GC-content, and sequence duplication level of the clean data were calculated. All the downstream analyses were based on clean data of high quality. The left files (read1 files) from all libraries/samples were pooled into one big left.fq file, and right files (read2 files) into one big right.fq file. Transcriptome assembly was accomplished based on the left.fq and right.fq files using Trinity, with min_kmer_cov set to 2 by default and all other parameters set to the default. Gene function was annotated based on the following databases: NR (NCBI non-redundant protein sequences); Pfam (protein family); KOG/COG/egg NOG (clusters of orthologous groups of proteins); Swiss-Prot (a manually annotated and reviewed protein sequence database); KEGG (the Kyoto Encyclopedia of Genes and Genomes); and GO (gene ontology). KOBAS 3.0 software was used to test the statistical enrichment of differential expression genes in the KEGG pathways.

### 2.8. Real-Time PCR

Liver tissue RNA was extracted using Trizol, chloroform, isopropanol, and ethanol, and its concentration and purity were assessed. The samples for the quantitative polymerase chain reaction (qPCR) were prepared by first performing the reverse transcription of RNA, then carrying out amplification with 35 cycles, using a two-step method. The ΔCt method was used to analyze the qPCR data. The primers are listed in Table 1.

### 2.9. Western Blot Analysis

Protein was extracted from the chipmunk livers using a lysis buffer, and the resulting lysates were centrifuged for 15 min at 4 °C. Subsequently, 15 μg of protein from each sample was subjected to sodium dodecyl sulfate-polyacrylamide gel electrophoresis (SDS-PAGE) and was then transferred onto a polyvinylidene difluoride (PVDF) membrane (Millipore, Bedford, MA, USA). The membrane was blocked with Tris-buffered saline (TBS) containing 5% nonfat dry milk for 1 h at room temperature, followed by incubation with the target primary antibodies (*CYP2C18*, 1:1000; *CYP26A1*, 1:1000; *Cpt1a*, 1:1000, and GAPDH, 1:2000) overnight at 4 °C. After washing with TBS-T (Tris-buffered saline with Tween 20), the membrane was incubated with the secondary antibody for 1 h at room temperature. The protein bands were visualized using an enhanced chemiluminescence (ECL) detection system, and the intensity of each band was quantified using Image J software (v1.54, Bethesda, MD, USA).

### 2.10. Statistical Analysis

SPSS 24.0 and R 4.0.5 software were used for data processing and statistical analysis. For measurement data, those variables following the normal distribution were expressed as mean and standard deviation, and Student’s *t*-test was used for comparisons between the groups. Variables not following a normal distribution were expressed as M (P_25_, P_75_), and comparison between groups was analyzed using the Mann–Whitney test. The data in this study followed the normal distribution, and the *t*-test was used for comparisons between groups.

## 3. Results

### 3.1. Effects of Hibernation on Liver Morphology of Chipmunk

The liver is a crucial organ for energy storage and metabolic processes. Figure 1A shows the overall appearance of liver samples from the hibernation group and the control group. The size and the weight were measured, and the liver coefficient was also calculated (Figure 1C,D). Comparison of the size and weight demonstrated that the hibernating chipmunks possessed larger liver sizes in terms of both volume and weight when compared to the normal chipmunks. As illustrated in Figure 1B, the liver of the hibernation group showed watery degeneration compared to the control group, but no inflammatory changes were observed.

### 3.2. The Energy Source of Chipmunk Livers During Hibernation

ATP levels in the livers of the chipmunks were analyzed with the ATP detection kit. The result indicated that ATP levels were higher in the livers of chipmunks in the hibernation group compared to the control group (Figure 2A), suggesting that chipmunks increase their ATP productivity in response to hibernation. As shown in Figure 2B, the glycogen was mostly stored in the liver and lungs in the control group, which was aligned with the dynamic activity of body temperature maintenance and breathing movement. Similar to the control group, glycogen was also mainly stored in the liver and lungs in the hibernation group. Interestingly, there was a notable increase in the level of hepatic glycogen, indicating that glucose might not be the main energy source in hibernation. This finding can be further verified by the blood glucose measurement outcome that, under the conditions of hibernation, the glucose level was significantly higher than in the control group (Figure 2C).

### 3.3. Proportion of Fatty Acids in the Liver During Hibernation

The result revealed that the genes involved in unsaturated fatty acid metabolism in chipmunks were significantly increased in the hibernation group, suggesting an ascending level of unsaturated fatty acids in the hibernating chipmunks’ livers (Figure 3A). Furthermore, the details of fatty acid contents in the chipmunks were measured by means of GC (gas chromatography) analysis among the hibernation groups (Figure 3B).

The result showed that the chipmunks in the hibernation group displayed a significant decrease in C18 fatty acid while showing an increase in C16, C20, and C22 fatty acids, in comparison with the control group (Figure 3B). Further analysis (Figure 3C) showed that the polysaturated fatty acid C18 decreased significantly in the hibernation group, while C15, C16, C20, and C22 increased significantly. As shown in Figure 3D, the levels of seven types of unsaturated fatty acids in the hibernation group increased, especially C20:3n6, C20:4n6, C20:5n3, and C22:6n3.

### 3.4. Transcriptome Analysis of the Liver of Hibernating Chipmunks

To further investigate the hepatic changes in function in chipmunks during hibernation, transcriptome sequencing and bioinformatics analysis were utilized. A total of 41.69 Gb of clean data were generated from the raw data, with RNA sequencing conducted after adequate quality control; over 20 million reads were obtained from both the control and hibernation groups. Subsequently, 50,361 unigenes were assembled from the sequenced reads. These unigenes were further annotated by comparing them with various databases such as COG and KEGG, to elucidate their respective gene functions.

The preliminary analysis of the differentially expressed genes revealed a total of 288 genes, comprising 133 upregulated genes and 155 downregulated genes (Figure 4B). Subsequently, the cluster of orthologous groups of proteins (COG) function classification analysis (Figure 4C) revealed significant alterations in the transport and metabolic functions of hepatic lipids during hibernation. Additionally, signal transduction mechanisms, carbohydrate transport and metabolism, post-translational modifications, protein turnover, and chaperones were also found to be significantly altered.

Based on the RNA-seq data of chipmunk livers, the KEGG classification annotation analysis was conducted to investigate the regulating pathways of the differentially expressed genes. Figure 5A depicts the annotation results of the differentially expressed genes categorized by pathway types in KEGG, which revealed significant differences in the expression of three types of hepatic lipid metabolism genes, arachidonic acid metabolism, and fatty acid metabolism. The results also indicated that the PPAR pathway possesses the most genes implicated in metabolism. Furthermore, the most prevalent environmental signaling pathways identified were the PI3K-AKT signaling pathway and the AMPK signaling pathway.

KEGG pathway enrichment analysis was employed to identify those pathways with significant enrichment according to the differentially expressed genes. As a result, 20 enriched signaling pathways (Figure 5B) were identified, and several upregulated differential genes were notably enriched in the retinol metabolism pathway, PPAR signaling pathway, and fatty acid metabolism pathway. What aroused our interest was that the first five signaling pathways are closely related to lipid metabolism, which suggests that lipid metabolism could serve as the prominent energy source in hibernation.

### 3.5. The Effect of Hibernation on Lipid Metabolic Genes Expression in the Liver of Chipmunks

Retinol is involved in multiple biological pathways. Figure 5B also demonstrates that the retinol signaling pathway was altered during hibernation in chipmunks. As shown in Table 2, most genes engaged in retinol metabolism belong to the cytochrome P450 family (CYP family). To validate its role in hibernation, qPCR was exploited to analyze several genes related to retinol metabolism. The outcome proved to be in agreement with the RNA-seq data, except for *CYP26A1* upregulation, where another CYP family was downregulated (Figure 6A). Western blot analysis was performed on the cytochrome proteins of *CYP26A1* and *CYP2C18*. The results are displayed in Figure 6C,D. In contrast to the control group, the expression of *CYP2C18* decreased, and *CYP26A1* expression increased in the hibernation group.

The PPAR pathway is mainly engaged in regulating lipid and glucose metabolism, playing a vital role in the maintenance of metabolic homeostasis. Transcriptome data about the hibernation group and control group were showed in Table 3. The genes related to PPAR pathway were also tested by qPCR. As shown in Figure 6B, the gene expression is consistent with Transcriptome data. *Cpt1a*, a gene involved in PPAR pathway, was chosen and its protein expression level was estimated by Western Blot (Figure 6C,D). The hibernation group exhibited a significant reduction in *Cpt1a* protein levels compared to the normal group, consistent with the findings from RNA sequencing.

## 4. Discussion

Hibernation is a special living adaptation in the face of extreme cold for mammals like squirrels and chipmunks in the smaller species and brown bears in the larger species [11]. Before hibernation, these animals consume a great deal of food to accumulate enough energy reserves to endure cold conditions throughout the winter [12]. The accumulation of adipose tissue is one of the characteristics of the necessary metabolic changes, along with those changes in morphology that we observed in our experiment. During hibernation, chipmunks maintain a relatively lower temperature due to reduced activity and metabolic reprogramming. Previous research has identified superior metabolic pathways that transition to fatty acid oxidation [13], suggesting that metabolic reprogramming happens in the liver under hypothermia stimulation. In this study, we carefully evaluated the ATP levels, blood glucose levels, and glycogen levels found in chipmunks during hibernation. The blood glucose levels and liver glycogen levels of the hibernating group of chipmunks were much higher than those of the control group, suggesting that glucose metabolism was not the main energy source for chipmunks during hibernation. Meanwhile, the cells of the hibernating group exhibited slight hydropic degeneration, which may result from alterations in the pathways or rate of energy generation within hibernating tissues. The sodium-potassium-ATP pumps on the cell membranes can effectively maintain stable osmotic pressure under normal conditions [14]. Prior research has already indicated that the enhanced tricarboxylic acid cycle is crucial to sustaining this energy balance under low-temperature conditions [15]. In our study, we demonstrated that the levels of glycogen, the most common energy reserves, were notably elevated in multiple tissues (such as the liver, heart, kidney, lung, and muscle) in the hibernation group under low-temperature conditions, which excluded the dominant role of glucose metabolism on the response to low-temperature conditions. Therefore, we examined the lipid metabolic levels, another vital source of the tricarboxylic acid (TCA) cycle substrate [16]. The enhanced expressed genes related to unsaturated fatty acid metabolism (ELOVL1, FADS, and SCD) and revealed that fatty acid metabolism played a crucial role in supporting the heightened TCA cycle activity necessary for maintaining energy homeostasis. Furthermore, it has been proposed that insulin resistance may serve as a key factor in metabolic reprogramming during hibernation [17]. Decreased sensitivity to insulin led to a decline in glucose consumption and storage as glycogen, which was consistent with our experimental outcome that low-temperature livers possessed a higher glucose level, while glycogen levels increased in the liver during hibernation. The slower and more subtle respiration rate and lower heartbeat may also contribute to decreased glucose metabolism, due to mild hypoxia.

Lipid metabolism is essential for both body temperature regulation and physiological function maintenance in hibernating animals [18]. Exposure to low temperatures could also activate the processing of brown adipose tissue (BAT) to enhance the tricarboxylic acid (TCA) cycle by promoting the utilization of fatty acids [19]. In accordance with our result, saturated fatty acids, for instance C18, decreased notably while unsaturated fatty acids, including C15, C16, C20, and C22, increased during hibernation. Some unsaturated fatty acid types displayed significant alteration, for example, C20:3n6, C20:4n6, C20:5n3, and C22:6n3. This conversion of fatty acids during hibernation could not only be an adaptation to cold stress to provide more energy but also help maintain homeostasis in the internal environment and increase cell viability as well. Under cold conditions, saturated fatty acids like C18, the level of which decreased during hibernation, might lead to changes in the cellular membrane component, resulting in a reduction in membrane fluidity and the activity of membrane-associated enzymes and receptors [20]. However, some saturated fatty acids, like palmitic acid (C16), increased during hibernation in order to inhibit the secretion of pro-inflammatory cytokines and regulate immunity. Palmitic acid can induce the palmitoylation of mitochondrial antiviral signaling proteins and subsequently promote the aggregation of mitochondrial antiviral signaling proteins [21]. Recently, scientists found that C15, with increasing levels of saturated fatty acid during hibernation, could also play an anti-inflammatory and protective barrier role by means of FATP4 and the stimulation of NF-κB, subsequently reducing proinflammatory cytokine mRNA expression [22].

In comparison with saturated fatty acids, the unsaturated fatty acids display similar effects to maintain homeostasis. For instance, C20:4n6, C20:5n3, and C22:6n3, apart from their crucial role in maintaining biofilm fluidity, are essential for regulating the cellular inflammatory response and promoting energy metabolism during hibernation [23]. Researchers have already found that the enzymatic conversion of saturated fatty acids to unsaturated fatty acids by lipid desaturase increased unsaturated fatty acids within membrane phospholipids, thereby enhancing membrane fluidity, alleviating the inflammation response, and mitigating the membrane sclerosis induced by low temperatures to maintain internal environment homeostasis and normal cellular function, which was consistent with our results of elevated unsaturated fatty acid levels in hibernating chipmunks’ livers.

This study also provided evidence of the involvement of retinol metabolism in lipid metabolism in the liver. The result of transcriptomics analyses in the chipmunk liver samples demonstrated that the gene expression level of retinol metabolism rose notably, according to the KEGG gene enrichment analysis. The ascending expression on a protein level was also verified in our experiment through Western blot analysis. The liver serves as the primary site for retinol storage, with approximately 80% of retinol stored as retinyl esters within lipid droplets in the hepatic stellate cells (HSC). Retinols, such as vitamin A (VA), all-trans retinol (ROL), and their analogs, are essential in many biological pathways [24], particularly in energy metabolism through the activation of the retinoic acid receptors RARα, RARβ, and RARγ [25]. Numerous studies have proved the correlation between decreased hepatic retinol levels and various liver diseases. In adult rats, a deficiency in vitamin A leads to the vacuolization and steatosis of liver cells [26]. Hepatic vitamin A levels are significantly reduced in non-alcoholic fatty liver disease (NAFLD), and there is a negative correlation between hepatic retinol levels and the severity of hepatic steatosis in both mice and humans with NAFLD [27,28]. Retinol can also upregulate FGF21 through retinol binding protein 4 (RBP4) and its rapid mobilization, facilitating lipid metabolism and improving insulin sensitivity [29]. This result suggests that hepatic cells might initiate a similar pattern under hypothermia conditions, which will subsequently enhance the retinol metabolism pathway and sensitivity to insulin, which will then result in the accumulation of glycogen and metabolic reprogramming.

During hibernation, alterations in hepatic lipid transport and metabolism in the chipmunks were identified through RNA-sequencing and bioinformatics analyses. The retinol metabolism pathway is highly engaged in the liver’s metabolic transition in hibernating chipmunks, with most of the genes belonging to the cytochrome P450 family. These are common in a variety of substance metabolism processes [30]: (1) Detoxification process: the conversion of drugs from hydrophobic to hydrophilic forms to facilitate their excretion. (2) Redox reactions: the monooxygenase enzyme P450 is involved in the metabolism of various substances. (3) Lipid metabolism promotion: P450 enzymes form a solubilizing medium with the help of lipids and thereby activate lipid metabolism. In addition, it was reported that P450 proteins possess the ability to trigger a vasodilatory effect for regulating tissue blood flow [31], which is associated with thermoregulation during hibernation. Some previous research studies have focused on the involvement of PPARα in metabolic regulation, for PPARα transcriptionally controls most of the enzymes responsible for mitochondrial uptake and fatty acid oxidative catabolism [32]. PPARα is commonly upregulated, along with PGC1-α, a cold-inducible transcriptional coactivator [33]. In our study, cpt1a shows a lower expression and is reported to be associated with long-chain fatty acid oxidation.

In conclusion, there are several reasons for energy supply mode conversion from glucose metabolism to fat-derived ketone body metabolism in the liver of chipmunks during hibernation. Firstly, it can avoid damage to tissue cells due to the low temperatures during hibernation. Secondly, it can avoid tissue damage from lactic acidosis, which is caused by hypoxia. Thirdly, the switch to lipid metabolism in terms of the energy supply is conducive to improving the animal’s ability to obtain higher levels of ATP and adapt to low temperatures. What inspired us is that research into hibernating animals can offer valuable insights into the impact of cold environments on metabolic processes and provide inspiration for therapeutic interventions to treat organic diseases. For instance, therapies based on lipid metabolism, such as a ketogenic diet, have already been utilized in the treatment of many diseases like cancer. In the future, more promising therapeutic approaches concerned with exposure to low temperatures will be developed, not only for metabolic disorders but for cancer as well. The subjects of this study were wild chipmunks, but there are many other types of mammals that hibernate, such as bears, bats, and some other rodents. In future studies, the types of subjects can be expanded to better explore the metabolic mechanisms of organisms during hibernation and help in the treatment of related diseases.

## 5. Conclusions

In conclusion, the energy supply pattern of chipmunks via the liver changes from glucose metabolism to fat metabolism during hibernation, and the signaling pathways related to liver lipid metabolism, such as the retinol signaling pathway and PPAR signaling pathway, are also changed. This plays an important role in coping with the challenges of hibernation.

## Figures and Tables

**Figure 1 animals-15-02091-f001:**
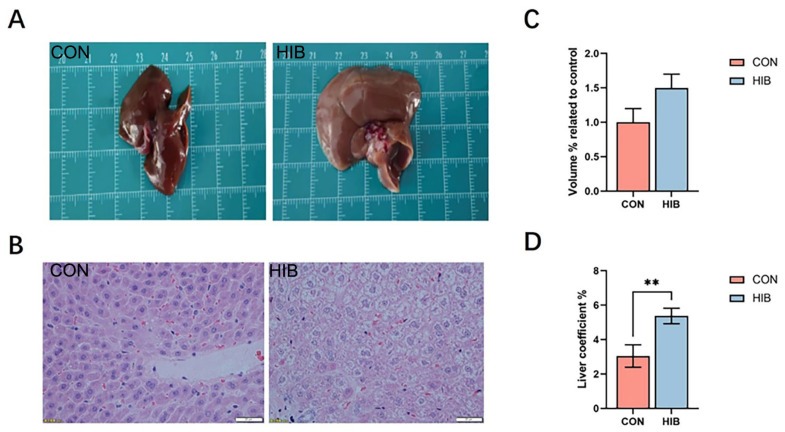
The different appearance and micrograms of liver samples between animals in hibernation and under normal conditions in chipmunks. (**A**): The gross appearance of liver samples from the hibernation group and control group; (**B**): the microscopic appearance of liver from the hibernation group and control group. Scale bars = 20 μm. (**C**): The volume of liver in the hibernation group and control group; (**D**): The liver coefficient in the hibernation group and in the control group. Data are presented as mean ± SD; *n* = 10. ** *p* < 0.01.

**Figure 2 animals-15-02091-f002:**
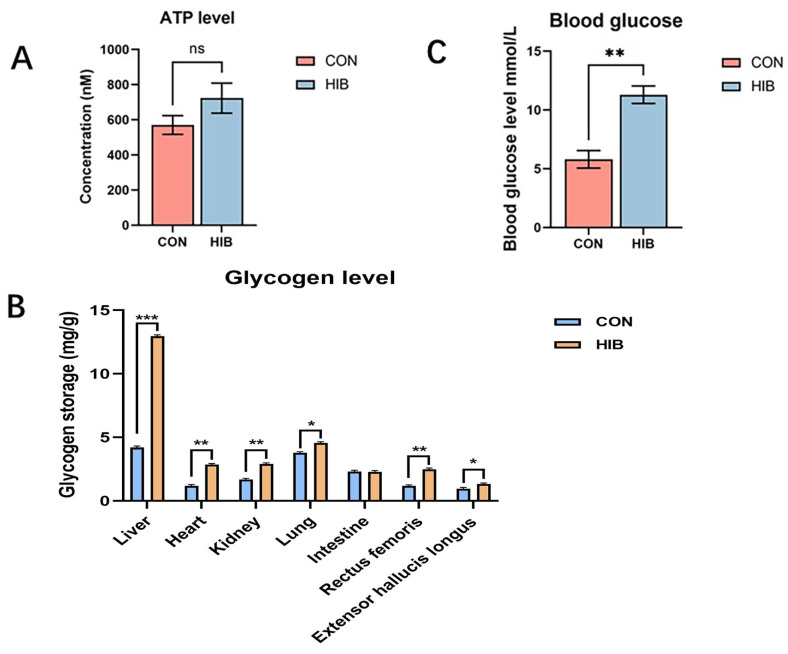
The ATP levels and glucose-related index between the hibernation and control groups. (**A**): ATP levels in the hibernation group and the control group; (**B**): the glycogen levels between the hibernation group and the control group in different organs of chipmunks; (**C**): the blood glucose levels between the hibernation group and the control group. Data are presented as mean ± SD, *n* = 10. * *p* < 0.05, ** *p* < 0.01, *** *p* < 0.001, ns = not significant.

**Figure 3 animals-15-02091-f003:**
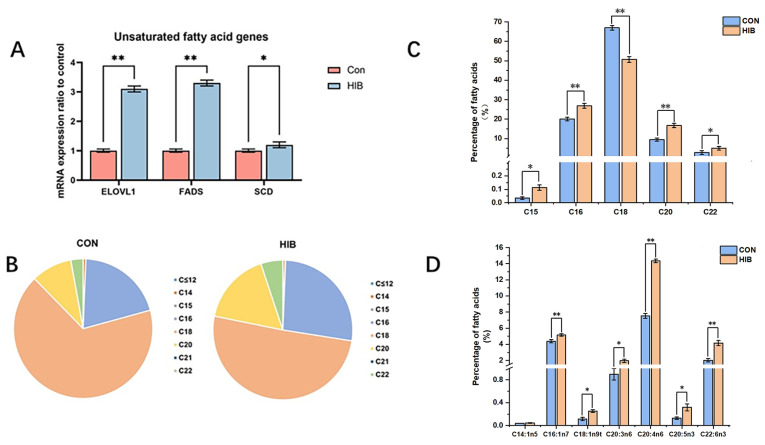
The expression level of lipid metabolism-related genes was detected by qPCR. (**A**): The common unsaturated fatty acid gene expression levels in the hibernation group and the control group. (**B**): The fatty acid content distribution diagram in the hibernation group and the control group. (**C**): The percentage composition of different fatty acids between the hibernation group and the control group. (**D**): The further classification of relatively significant polyunsaturated fatty acid and their percentage composition between the hibernation group and the control group. Data are presented as mean ± SD, *n* = 10. * *p* < 0.05, ** *p* < 0.01.

**Figure 4 animals-15-02091-f004:**
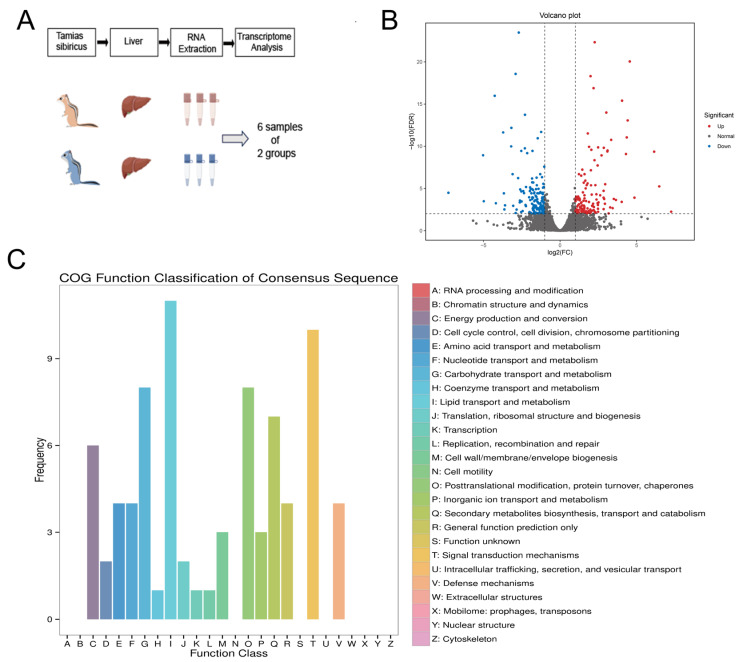
The transcriptome sequencing and bioinformatics analysis results. (**A**): RNA-seq experiment flow chart in chipmunks. (**B**): Volcano diagram showing those genes that are differentially expressed in hibernating mice. (**C**): COG function classification analysis indicating an evident transition to lipid metabolism in hibernating mouse liver.

**Figure 5 animals-15-02091-f005:**
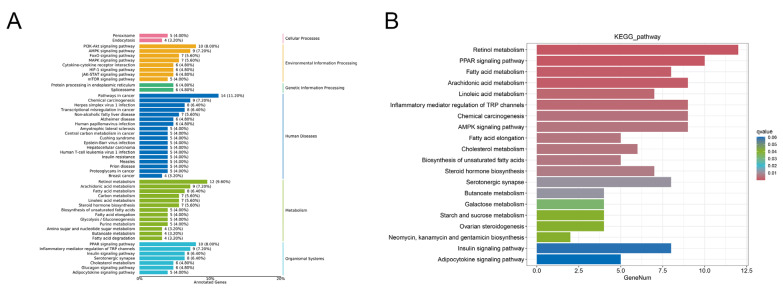
The KEGG classification annotation and KEGG pathway enrichment analysis results. (**A**): Fatty acid transport and metabolism pathways were enhanced in accordance with KEGG classification annotation. (**B**): KEGG pathway enrichment analysis displays the number of genes in different enrichment pathways, with a notable increase in the retinol metabolism pathway.

**Figure 6 animals-15-02091-f006:**
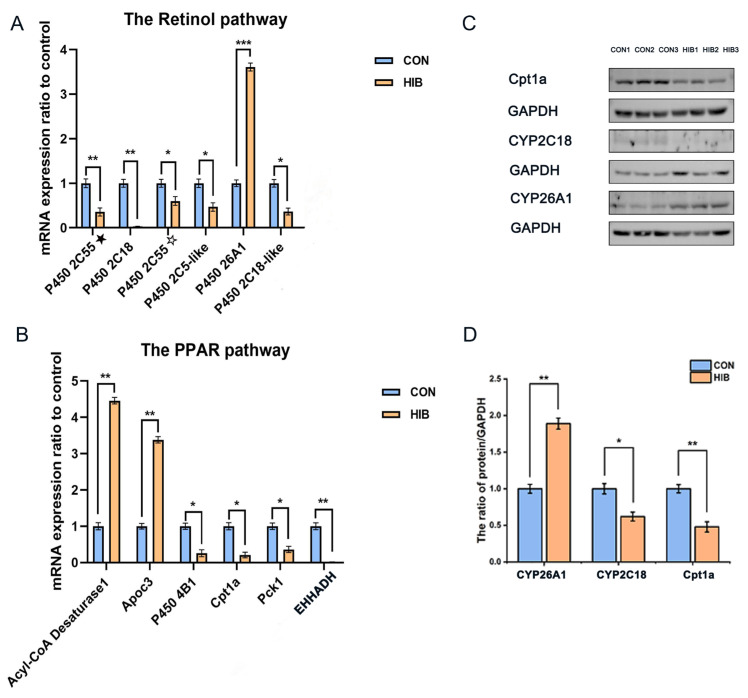
The mRNA and protein expression levels recorded during hibernation. (**A**): The mRNA expression ratio in the retinol pathway. (**B**): The mRNA expression ratio in the PPAR pathway. (**C**): The protein expression level of the *CYP26A1*, *CYP2C18*, *Cpt1a* and GAPDH. (**D**): The quantification of protein expression. Data are presented as mean ± SD, *n* = 10. * *p* < 0.05, ** *p* < 0.01, *** *p* < 0.001. *P450 2C55*★: *P450 2C55*(*LOC124096538*), *P450 2C55*☆: *P450 2C55*(*LOC101972885*).

**Table 1 animals-15-02091-t001:** The primers identified in the study.

Primer Name	Sequence (5′–3′)
*P450 2C55* (*LOC124096538*)	CAACTCTCCCTCTTTCTGC
	ACACACCACACTCATCCTG
*P450 2C18* (*LOC124096544*)	CTTATCCCAACCAATCTGC
	TGAGTCCATCAAGGTGCT
*P450 2C55* (*LOC101972885*)	TGTCTGCTTCTTGTCTCACTC
	ATCATCCAGGGCTTCCTT
*P450 2C5-like* (*LOC101972028*)	TTTCCTCAACTCCTCCACC
	ACAAGACGCTTCTCCCTCA
*P450 26A1* (*LOC107154487*)	GATGACCCGCAATCTCTT
	ACGAGTGCTCAATCAACAG
*P450 2C18-like* (*LOC101976849*)	TGTCCAAAGAAGAGCAGTG
	GGTTGATGATAAGGAGAGCA
*Acyl-CoA Desaturase1* (*LOC124984655*)	GGTTATTTGGGTAGTTGGC
	TCACTGACCTGGGATTGT
*Apoc3*	CCCTGACTTCACACATCTCC
	TGCTACCCACTCTCCTTCAC
*P450 4B1* (*LOC124963612*)	GGTAAAGGAGAGGAGGCAA
	AACGACACCATCTACGGTC
*Cpt1a*	AAACCCACCAGGCTACAGTG
	GCAGGTCCACATCATTTGC
*PCK1*	TCTCGGGTGATGATGACT
	GAAATGTGACCAGGAGTGA
*EHHADH*	CCATTGCCACTGTTATGAAC
	GCTTGCTGCCTTCTTCTAA
*GAPDH*	GTGATGCTGGTGCTGAATA
	GCTGACAATCTTGAGGGA

**Table 2 animals-15-02091-t002:** Retinol metabolism-related genes.

Gene Name	Con	Hib	Ratio	Ratio (Up/Down)
Marmota monax cytochrome *P450 2C55* (*LOC124096538*)	444.8867	117.8467	3.775132	−3.78
Marmota monax cytochrome *P450 2C18* (*LOC124096544*)	439.3067	46.88	9.370876	−9.37
Ictidomys tridecemlineatus cytochrome *P450 2C55* (*LOC101972885*)	79.36333	16.56667	4.790543	−4.79
Ictidomys tridecemlineatus cytochrome *P450 2C5-like* (*LOC101972028*)	239.2967	74.70667	3.20315	−3.20
Marmota monax retinol dehydrogenase 16 (*Rdh16*)	107.9867	50.68667	2.130475	−2.13
Marmota cytochrome *P450 26A1* (*LOC107154487*)	1.173333	7.206667	6.142045	6.14
Ictidomys tridecemlineatus cytochrome *P450 2C18-like* (*LOC101976849*)	639.6767	116.9067	5.471687	−5.47
Ictidomys tridecemlineatus cytochrome *P450 2C18-like* (*LOC101976849*)	809.18	443.1967	1.825781	−1.83
Marmota 17-beta-Hydroxysteroid dehydrogenase *13*(*LOC107153606*)	443.02	211.03	2.099322	−2.10
Marmota monax cytochrome *P450 2C5* (*LOC124096537*)	400.53	113.7567	3.520937	−3.52
Marmota monax cytochrome *P450 2C18* (*LOC124096535*)	112.2567	50.06667	2.242144	−2.24
Marmota cytochrome *P450 2C18-like* (*LOC107149742*)	433.4967	44.40333	9.762706	−9.76

**Table 3 animals-15-02091-t003:** PPAR signaling pathway-related genes.

Gene Name	Con	Hib	Ratio	Ratio (Up/Down)
Sciurus carolinensis acyl-CoA Desaturase1 (*LOC124984655*)	30.17	96.65	3.203513	3.20
Ictidomys tridecemlineatus apolipoprotein C3 (*Apoc3*)	2.723333	13.03	4.784578	4.78
Ictidomys tridecemlineatus glycerol kinase (*Gk*)	46.00667	23.46	1.961068	−1.96
Sciurus carolinensis cytochrome P450 4B1 (*LOC124963612*)	444.8867	117.8467	3.775132	−3.78
Marmota monax enoyl-CoA hydratase and 3-hydroxyacyl CoA dehydrogenase (*Ehhadh*)	526.05	91.17333	5.769779	−5.77
Urocitellus parryii angiopoietin-like 4 (Angptl4)	100.91	27.60333	3.655718	−3.66
Marmota carnitine palmitoyltransferase 1A (*Cpt1a*)	92.10333	33.17667	2.776148	−2.78
Marmota monax phosphoenolpyruvate carboxykinase1 (*Pck1*), transcript variant X1	758.3533	76.38333	9.928257	−9.93
Marmota flaviventris 3-hydroxy-3-methylglutaryl-CoA synthase 2 (*Hmgcs2*)	1303.4	579.67	2.248521	−2.25
Marmota monax 7-alpha-hydroxycholest-4-en-3-one 12-alpha-hydroxylase (*LOC124087568*), transcript variant X1	106.2467	3.553333	29.90056	−29.90

## Data Availability

The study’s original contributions are included in the article. Further inquiries can be directed to the corresponding authors.

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
