# Peer review of "The Role of Lipid Metabolic Reprogramming in the Hibernation of Chipmunks"

_animals, 2025, doi:10.3390/ani15142091_

Round 1

Reviewer 1 Report

Comments and Suggestions for Authors

By comparing lipid-metabolism differences between hibernating and active chipmunks, the authors aim to reveal how animals adapt to low temperatures and to link these findings to diseases caused by human hepatic metabolic disorders. The study’s angle is scientifically sound and engaging; although mammalian hibernation biology is already well explored, this work still adds to the field. The results are reliable and thorough, but the Materials and Methods section needs a deeper, more detailed revision—this is crucial for ensuring reproducibility. I believe that the study is suitable for publication in Animals, but still offers some suggestions for improvement in certain parts of the manuscript.

Line specific comments/questions:

L 20: “of” to “on”.

L 23: “including” to “by”

L 36: Protein isn’t a short-term energy source.

L 42: Please provide some relevant information on chipmunks’ hibernation habits.

L 77: During the experiment, were the chipmunks in the control group given food, and if so, how often? Were the animals in the experimental group fed as well? Why was the environment kept dark? In reality, when animals hibernate they build nests in sun-exposed spots to increase the temperature.”

L 97: A detailed description of the glycogen assay procedure should be provided here.

L 100: First, the sample-collection details should be merged into the earlier liver-sampling section; second, the exact time points for blood draws must be specified; and finally, how was the blood handled after collection—did you isolate whole blood, plasma, or serum?

L 108: So, how exactly did you measure that final ATP—what kit did you use, which instrument, and how were the results presented?

L 110: Please double-check this sentence—the experimental animals are all wrong. And which sample did you analyze: blood or liver?

L 122-131: The transcriptome-sequencing methods and bioinformatics section need to be rewritten—the present description is far too brief. Specifically, the bioinformatics analysis never states which software was used for QC, filtering, alignment, and annotation. The paper also doesn’t clarify whether you performed reference-guided or de novo transcriptomics; if it’s reference-guided, which genome did you use as the reference?

L 133: Which genes did you quantify by qPCR, and what primers did you use for each target gene and for the internal reference gene?

L 151: First, not every dataset is suited to “mean ± SD” presentation, so the authors should state each variable explicitly (e.g., glycogen levels, relative gene expression, etc.). Second, data must be tested for normality before applying a t-test; if normality is violated, a non-parametric Mann–Whitney U test is required.

L 156: Why did body-weight data suddenly appear? The timing and method of this measurement aren’t mentioned at all in the Materials and Methods section.

L 176: The Materials and Methods section only mentions collecting liver and blood samples, whereas the Results imply that many other sample types were also obtained. Please provide a more detailed description of all sample-collection procedures in the Materials and Methods section.

L 518: check this reference

Reviewer 2 Report

Comments and Suggestions for Authors

The manuscript entitled “The role of lipid metabolic reprogramming in hibernation of chipmunks” is original and interesting, particularly due to its relevance to human metabolic diseases, which makes the work potentially valuable for translational medicine as well.
However, several limitations currently prevent the manuscript from being suitable for immediate publication. Below, I provide some suggestions for improvement.

The method of euthanasia is unclear, and the description provided is not sufficiently detailed to dispel methodological and ethical concerns. A more thorough explanation of the protocol followed is necessary, including a reference to an ethically recognized method or guideline.

  1. 110: Why is there a reference to “fetal mouse brain tissue”? Please clarify its relevance or consider removing it if not applicable.

L.120: You state that fatty acid analysis was performed by GC (Gas Chromatography), whereas in L. 192, you mention HPLC (High Performance Liquid Chromatography). This appears to be inconsistent: could you please clarify the methodology used?

  1. 190: Instead of “fatty acid genes,” it would be more accurate to say “genes involved in fatty acid metabolism.”

The Discussion section is overly long and somewhat repetitive, particularly regarding the description of lipid and glucose metabolism. I recommend avoiding unnecessary repetition and improving the logical flow and cohesion of the narrative.

In the Conclusions, a strong final summary is lacking. Most importantly, the limitations of the study and suggestions for future research are not addressed and should be included.

Overall, the English language should be improved to enhance fluency and clarity throughout the manuscript.

Comments on the Quality of English Language

Overall, the English language should be improved to enhance fluency and clarity throughout the manuscript.

Round 2

Reviewer 1 Report

Comments and Suggestions for Authors

The authors have provided thorough responses to my review comments, and I acknowledge the satisfactory revisions made to the manuscript.

Author Response

Thank you for taking the time to review our manuscript. We appreciate your valuable feedback and constructive suggestions. Thank you for your hard work on our manuscript. Wish you a happy life and smooth career!

Reviewer 2 Report

Comments and Suggestions for Authors

The authors have responded promptly to every observation made. There are a few points that need clarification, listed below.

L. 322-323: The sentence: "The hibernation group showed a much higher level ..." is incomplete. It should be specified what the much higher level refers to.

L. 345: “dispensable” should be “essential”?

L. 356: What do you mean by "biofilm fluidity"? Perhaps "membrane fluidity"?

I would recommend a thorough proofreading by a native speaker.

Comments on the Quality of English Language

I would recommend a thorough proofreading by a native speaker.
